# Aberrant Lymphatic Drainage in the Contralateral Axilla in Patients with Isolated Ipsilateral Breast Tumor Recurrence

**DOI:** 10.3390/jcm9041192

**Published:** 2020-04-22

**Authors:** Jai Min Ryu, Byung Joo Chae, Jeong Eon Lee, Jonghan Yu, Seok Jin Nam, Seok Won Kim, Se Kyung Lee

**Affiliations:** Division of Breast Surgery, Department of Surgery, Samsung Medical Center, Sungkyunkwan University School of Medicine, Seoul 06351, Korea; jaimin.ryu@samsung.com (J.M.R.); bjchae@gmail.com (B.J.C.); jeongeon.lee@samsung.com (J.E.L.); lymbics@hanmail.net (J.Y.); seokjin.nam@samsung.com (S.J.N.); seokwon1.kim@samsung.com (S.W.K.)

**Keywords:** Breast Neoplasm, Sentinel Lymph Node Biopsy, Local Recurrence

## Abstract

The management and implications of aberrant lymphatic drainage in the contralateral axilla during repeat sentinel lymph node biopsy (SLNB) in patients with isolated ipsilateral breast tumor recurrence (IBTR) are not well understood. We analyzed the outcomes of contralateral SLNB in cases of isolated IBTR compared to ipsilateral SLNB. We conducted a retrospective review of cases reported at Samsung Medical Center between 1995 and 2015. All patients with isolated IBTR that underwent ipsilateral and contralateral SLNB with clinically negative lymph nodes but lymphatic drainage on the ipsilateral or contralateral axilla were included. Among 233 patients with isolated IBTR, 31 patients underwent repeat SLNB, 11 underwent ipsilateral SLNB, and nine underwent contralateral SLNB. None of the patients showed contralateral axillary metastasis in cases with isolated IBTR in the absence of clinically suspicious drainage on the contralateral axilla. Contralateral drainage was associated with a longer interval to IBTR (68.4 vs.18.6 months, *p* = 0.001) and the overall median follow-up duration (102.6 vs. 45.4 months, *p* = 0.002). There was no significant difference in the recurrence after the second operation (1 of 11 vs. 1 of 9, *p* = 1.000). Only one patient in both groups experienced recurrence after the second operation. Two patients (22.2%) who underwent contralateral SLNB had lymphedema. We demonstrate that no patient had contralateral metastasis in patients with isolated IBTR in the absence of clinically suspicious drainage in the contralateral axilla. Further study is warranted to better understand and optimize the management of these rare and challenging cases.

## 1. Introduction

Sentinel lymph node biopsy (SLNB) is the standard treatment for axillary lymph node staging in patients with clinically node-negative breast cancer, replacing axillary lymph node dissection (ALND) [1,2]. Although there is no significant difference in overall survival between breast-conserving surgery (BCS) following adjuvant radiotherapy (RTx) and total mastectomy (TM), BCS shows higher isolated ipsilateral breast tumor recurrence (IBTR) than TM [3,4,5,6]. There has been an increase in IBTR because BCS has become a standard breast surgery treatment and the proportion of BCS among breast surgery cases has increased [7]. Accordingly, repeat (re-) SLNB in recurrent breast cancer cases has also increased. Several studies have suggested that re-SLNB in recurrent breast cancer is feasible [8,9,10,11,12,13].

If the patients who had IBTR identified with contralateral axillary metastasis, it can be considered as stage IV. Therefore, contralateral SLNB could be important to avoid underestimation of the status of the disease. On the other hand, patients who underwent both axillary surgeries show decreased quality of life because they should check blood pressure or take blood samples at legs to prevent higher risk of lymphedema. However, very few reports have discussed the management and implications of aberrant lymphatic drainage after SLNB in patients with isolated IBTR, especially in the contralateral axilla. Here, we report factors for predicting contralateral drainage, and compare the prognosis between ipsilateral and contralateral SLNB groups in patients with isolated IBTR.

## 2. Materials and Methods

A retrospective review was conducted to identify all patients with isolated IBTR who underwent re-SLNB, including ipsilateral and contralateral SLNB, with clinically negative lymph nodes at the ipsilateral and contralateral axilla at Samsung Medical Center (SMC) between January 1995 and December 2015. Among 17,332 primary breast cancer patients were operated at SMC and 1032 patients with loco-regional recurrence were identified. Among them, 233 patients with isolated IBTR were identified and 31 patients that underwent re-SLNB. Isolated IBTR was defined as recurrent breast cancer within the ipsilateral breast. We reviewed the following clinicopathological characteristics and clinical outcomes of both the initial and the second operation: age, type of breast and axillary operation, tumor location, number of positive lymph nodes, number of retrieved lymph nodes, estrogen-receptor (ER) status, progesterone-receptor (PR) status, human epidermal growth factor-2 (HER-2) status, nuclear grade, pathologic stage based on the seventh American Joint Committee on Cancer classification, adjuvant treatment before and after operation, and SLNB detection method. We also reviewed the clinical course and lymphedema after the second operation.

### 2.1. SLNB Mapping Technique

We conducted lymphatic mapping using technetium-99m sulfur colloid diluted in normal saline solution and/or vital blue dye (1% indigo-carmine). The site and timing of the mapping agent administration was at the physician’s discretion. Radiolabeled colloid was injected 2 h to 6 h before surgery and/or 1–5 mL of 1% indigocarmine was injected peri-areolarly and/or peri-tumoraly and the breast was massaged for 3–5 min. Lymphoscintigraphy was performed 30 min to 6 h before surgery. A handheld gamma detection probe (NeoProbe2000; US Surgical, Norwalk, CT) was used to scan the axilla transcutaneously and to identify the most radioactive area. All radioactive and/or blue lymph nodes (LNs) and suspicious palpable LNs were excised and submitted as sentinel lymph nodes (SLNs). The dual method is defined as the use of 1% indigocarmine as well as radiolabeled colloid. This study adhered to the ethical tenets of the Declaration of Helsinki and was approved by the Institutional Review Board (IRB) of SMC (IRB number: 2017-10-046). The need for informed consent was waived because of the retrospective nature of this study.

### 2.2. Statistics

Patient characteristics were compared by Wilcoxon rank sum tests for continuous variables and Fisher’s exact tests for categorical variables. For all analyses, a *p* < 0.05 was considered statistically significant. All statistical analyses were performed by SAS 9.4 software (SAS Institute, Cary, NC, USA) and R 3.2.5 (R Foundation for Statistical Computing, Vienna, Austria; http://www.r-project.org/).

## 3. Results

Overall, of the 17,332 patients that underwent an operation for breast cancer at SMC between January 1995 and December 2015, an isolated IBTR occurred in 233 cases. None of the cases were clinically suspected for axillary lymph node metastasis, including of the contralateral axilla. Re-SLNB was attempted in 31 patients, 11 underwent ipsilateral SLNB, nine underwent contralateral SLNB due to contralateral axilla lymphatic drainage on lymphoscintigraphy, and 11 patients were designated as failed for re- SLNB because they did not uptake at lymphoscintigraphy on either axilla (Figure 1).

Basic characteristics of the patients who underwent ipsilateral and contralateral SLNB are summarized in Table 1. There was no significant difference in the type of initial breast surgery (BCS; 90.9% vs 88.9%, *p* = 1.000) and axillary surgery (SLNB; 90.9% vs 55.6, *p* = 0.166) between the ipsilateral and contralateral SLNB groups. In addition, there was no significant difference in the initial stage (*p* = 0.890), number of resected (*p* = 0.878), and positive lymph nodes (*p* = 0.110) between the ipsilateral and contralateral SLNB groups. Two patients (22.2%) developed lymphedema after contralateral SLNB.

Detailed characteristics of contralateral SLNB patients are summarized in Table 2. Three patients were stage II, five were stage I, and one was stage 0. Most patients (8/9, 88.9%) underwent adjuvant radiotherapy. The second breast surgery was a mastectomy in 6/9 cases (66.7%). All patients with recurrence were diagnosed in stage I. Four patients underwent only contralateral SLNB because they were showed involvement of only on the contralateral axilla not ipsilateral axilla on lymphoscintigraphy (Figure 2).

Among them, 3 (75.0%) underwent axillary lymph node dissection (ALND) as the initial surgery. A second ipsilateral axillary surgery was performed on 5/9 cases (55.6%); 4/5 of these cases (80.0%) did not show axillary lymph node metastasis. Five patients showed uptake at both axillaries and underwent Sentinel lymph node biopsy (SLNB) on both sites (Figure 3).

The clinical course is summarized in Table 3. Contralateral drainage was associated with a longer interval to IBTR (68.4 vs. 18.6 months, *p* = 0.001) and a longer overall median follow-up duration (102.6 vs. 45.4 months, *p* = 0.002). There was no significant difference in the follow up duration after the second operation (26.6 vs. 33.2 months, *p* = 0.703) and there was no significant difference in the recurrence after the second operation (1 of 11 vs. 1 of 9, *p* = 1.000). Only one patient in both groups experienced recurrence after the second operation.

## 4. Discussion

Management of axillary lymph nodes in patients with isolated IBTR traditionally followed two approaches: an axillary operation was not performed if patients underwent ALND during the initial operation, or ALND without re-SLBN was performed if patients underwent SLNB alone during the initial operation. Although several studies have demonstrated that re-SLNB is feasible in patients with isolated IBTR, there is currently no standard guideline for managing aberrant lymphatic drainage during re-SLNB. We reported detailed characteristics of contralateral SLNB among patients with isolated IBTR, and compared the results to ipsilateral SLNB. All patients who underwent SLNB showed clinically node-negative status at the contralateral axilla with uptake lymphoscintigraphy. There was no significant difference in the clinico-pathologic characteristics and recurrence after the second operation between the ipsilateral and contralateral SLNB groups. The contralateral SLNB group showed longer IBTR duration compared to the ipsilateral SLNB group, and two patients (22.2%) who underwent contralateral SLNB had lymphedema.

A few reports have discussed the contralateral axillary in re-SLNB patients with isolated IBTR [14,15,16,17]. Cordoba et al. [18] reported that 54 patients with IBTR received re-SLNB. Thirteen patients (30%) who previously received ALND showed contralateral axilla drainage, while none of the patients in the previous SLN group showed contralateral axilla drainage, and of those, only one patient had contralateral axillary metastasis. Uth et al. [17] reported a nationwide retrospective study that included twelve Departments of Breast Surgery in Denmark. Among the 147 patients with isolated IBTR who underwent re-SLNB, nine had aberrant drainage and one had contralateral axillary LN metastasis. Unfortunately, there was no detailed information on the locations of the aberrant LN drainage.

SLNB is considered to be safe and to have a low associated risk of lymphedema [19,20,21]. However, in some studies, lymphedema is still a non-negligible complication for patients who have undergone SLNB. A meta-analysis of 28 articles and 9588 patients found that the incidence of lymphedema after SLNB ranged from 0%–63.4% [22]. Furthermore, many countries recommend avoiding blood draws, vaccinations, and blood pressure readings if possible [23]. Patients who undergo contralateral SLNB are able to have blood drawn and blood pressure is measured only at the leg, therefore, this procedure may increase inconvenience and decrease the patient’s quality of life.

Contralateral axillary metastasis is considered distant metastasis, therefore, contralateral axillary drainage should not be underestimated [24]. However, the reported incidence of contralateral axillary lymph node metastasis is only between 3.5% and 6.0% in patients with breast cancer [25]. Even a nation-wide registration study, which was a Sentinel Node and Recurrent Breast Cancer (SNARB) study, could not analyze the factors that could affect the contralateral axilla metastasis [26]. They reported 536 patients with locally recurrent non-metastatic breast cancer. Of the 333 successfully visualized patients, 180 (54.1%) showed aberrant drainage. Among 57 cases (31.7%) were identified in the contralateral axilla, and 37 (20.6%) in the contralateral axilla in addition to the ipsilateral internal mammary, contralateral periclavicular, ipsilateral periclavicular, or ipsilateral axilla. Of the 287 patients with successful re-SLNB, 230 (80.1%) were node-negative, while 46 (16.0%) had LN metastasis. Among these, 12 had contralateral lymph node metastasis: four cases were micro-metastasis and eight cases were macro-metastasis. Unfortunately, there were no descriptions of the clinical lymph node status on the contralateral axilla.

There were some limitations to our study. First, because our data were collected retrospectively, there was no consensus or guidelines for managing the contralateral SLNB. Some of the patients with uptake on lymphoscintigraphy may not have undergone contralateral SLNB. Second, given that we identified only nine patients with no metastasis at the time of contralateral SLNB, additional analysis is needed. In the future, a nation-wide study is warranted to further investigate and define guidance. Third, it is not clear that IBTR is truly a recurrence or simply a new primary cancer in the same breast. Lastly, aberrant lymphatic drainage could be identified to the internal mammary, periclavicular lymph node, and contralateral axilla, but we demonstrated only contralateral axilla. Further multicenter studies are needed to evaluate the effect of aberrant lymphatic drainages pathways.

In conclusion, we demonstrated that there was no significant difference in the recurrence after the second operation between the ipsilateral and contralateral SLNB groups. In addition, no contralateral axillary metastasis was identified in patients with isolated IBTR in the absence of clinically suspicious metastasis on the contralateral axilla. Further studies are needed to better define and optimize management of these rare and challenging cases.

## Figures and Tables

**Figure 1 jcm-09-01192-f001:**
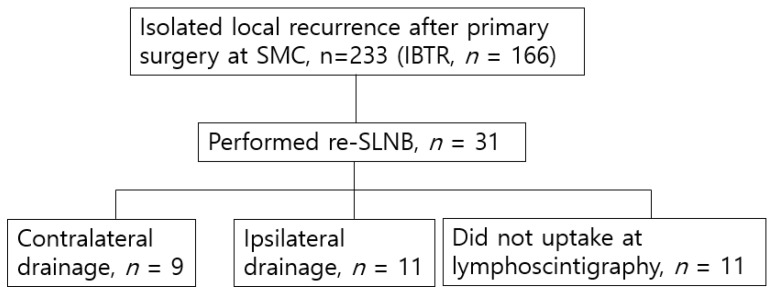
Schematic diagram for patient selection. SMC, Samsung Medical Center; IBTR, ipsilateral breast tumor recurrence; re-SLNB, re-sentinel lymph node biopsy.

**Figure 2 jcm-09-01192-f002:**
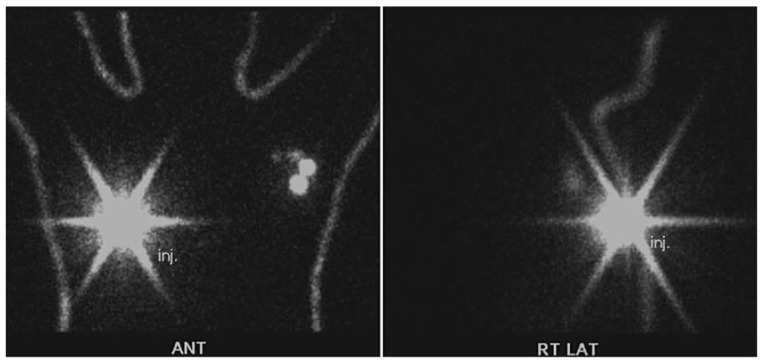
Aberrant drainage to the contralateral axilla and not on the ipsilateral axilla. ANT, anterior; RT, Right; LAT, lateral.

**Figure 3 jcm-09-01192-f003:**
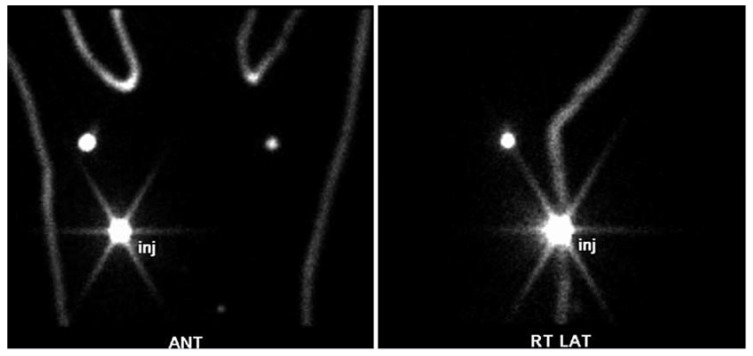
Aberrant drainage to the contralateral and ipsilateral axilla. One patient with axillary lymph node metastasis experienced recurrence on the flap after a skin-sparing mastectomy with a deep inferior epigastric perforator flap. Five patients underwent a dual method approach and four underwent radioisotope assessment only to detect the SLNs. The median number retrieved lymph nodes was 2.67 (range 1–4). None of the patients showed contralateral axillary metastasis in cases with isolated IBTR in the absence of clinically suspicious metastasis on the contralateral axilla.

**Table 1 jcm-09-01192-t001:** Basic characteristics of ipsilateral and contralateral Sentinel lymph node biopsy (SLNB) group.

	Ipsilateral SLNB	Contralateral SLNB	*p* Value
Number, *n*	11	9	
Median age at first operation, years, median (range)	47.00 (41.50, 49.50)	49.00 (40.00, 55.00)	0.493
Location			0.197
Right	5 (45.5)	7 (77.8)	
Left	6 (54.5)	2 (22.2)	
Primary tumor stage			0.890
0	1 (9.1)	1 (11.1)	
I	7 (63.6)	5 (55.6)	
II	2 (18.2)	3 (33.3)	
III	1 (9.1)	0 (0.0)	
Median age at second operation, years, median (range)	50.00 (42.50, 51.50)	51.00 (46.00, 58.00)	0.195
HR status			0.642
positive	7 (63.6)	7 (77.8)	
negative	4 (36.4)	2 (22.2)	
HER-2 status			0.670
amplification	6 (54.5)	6 (66.7)	
no amplification	5 (45.5)	3 (33.3)	
Primary breast surgery			1.000
BCS	10 (90.9)	8 (88.9)	
TM	1 (9.1)	1 (11.1)	
Primary axillary surgery			0.166
SLNB	10 (90.9)	5 (55.6)	
ALND	1 (9.1)	3 (33.3)	
No surgery	0 (0.0)	1 (11.1)	
Mean number of resected node (1st), number	7.09 (6.64)	8.22 (7.61)	0.878
Number of resected node (1st)			0.617
0–10	9 (81.8)	6 (66.7)	
11≥	2 (18.2)	3 (33.3)	
Mean number of positive node (1st), number	0.55 (1.81)	1.56 (2.96)	0.110
Number of positive node (1st)			0.074
0	10 (90.9)	6 (66.7)	
1–3	0 (0.0)	3 (33.3)	
4–9	1 (9.1)	0 (0.0)	
10≥	0 (0.0)	0 (0.0)	
Second breast surgery			0.033
BCS	0 (0.0)	4 (44.4)	
TM	10 (90.9)	5 (55.6)	
Unknown	1 (9.1)	0 (0.0)	
Second axillary surgery			0.211
SLNB	7 (63.6)	9 (100.0)	
ALND	3 (27.3)	0 (0.0)	
No axillary surgery	1 (9.1)	0 (0.0)	
Number of resected node (2nd), number	5.18 (4.77)	2.67 (0.87)	0.025
Number of resected node (2nd)			0.479
0–10	9 (81.8)	9 (100.0)	
11≥	2 (18.2)	0 (0.0)	
Mean number of positive node (2nd), number	0.45 (1.04)	0.00 (0.00)	0.215
Number of positive node (2nd)			0.479
0	9 (81.8)	9 (100.0)	
1–3	2 (18.2)	0 (0.0)	
Method of detection			0.642
Dual method	8 (72.7)	5 (55.6)	
Single method	3 (27.3)	4 (44.4)	
Primary chemotherapy			1.000
Yes	7 (63.6)	5 (55.6)	
No	4 (36.4)	4 (44.4)	
Primary radiotherapy			0.102
Yes	5 (45.5)	8 (88.9)	
No	4 (36.4)	1 (11.1)	
Unknown	2 (18.2)	0 (0)	
Interval to IBTR, months	18.6 (7.9)	68.4 (40.1)	0.001
Lymphedema after SLNB			0.891
Yes	2 (18.2)	2 (22.2)	
No	8 (72.7)	8 (88.9)	
Unknown	1 (9.1)	0 (0)	

HR: hormonal status; HER-2: human epidermal growth factor 2; BCS: breast conserving surgery; TM: total mastectomy; SLNB: sentinel lymph node biopsy; ALND: axillary lymph node dissection.

**Table 2 jcm-09-01192-t002:** Characteristics of patients who had isolated ipsilateral breast tumor recurrence (IBTR) and underwent contralateral sentinel lymph node biopsy.

Case No.	Age	Site	Operation	Histopathology	Np	Nd	Stage	ER	PR	HER-2	NG	Method of Detection	Location of Radioisotope Uptake	Adjuvant Tx.	Contralateral SLNB
**Initial operation**															
1	61	Rt	BCS with SLNB	IDC	0	2	I	Positive	Positive	Positive	High	Dual method	Rt axilla	CTx, RTx, HTx	Not performed
2	56	Rt	BCS with ALND	IDC	1	19	IIa	Negative	Negative	Negative	Intermediate	Dual method	Rt axilla	CTx, RTx	Not performed
3	50	Rt	BCS with ALND	IDC	1	14	IIa	Positive	Positive	Negative	Low	Radioisotope	Rt axilla	CTx, RTx, HTx	Not performed
4	44	Rt	BCS with ALND	IDC	3	20	IIb	Positive	Positive	Negative	Intermediate	Dual method	Rt axilla	CTx, RTx, HTx	Not performed
5	34	Lt	BCS with SLNB	IDC	0	4	ypI	Positive	Positive	Positive	Intermediate	Dual method	Lt axilla	NACT, RTx, HTx	Not performed
6	49	Rt	BCS with SLNB	IDC	0	3	I	Positive	Positive	Positive	Intermediate	Radioisotope	Rt axilla	RTx, HTx	Not performed
7	55	Rt	SSM with SLNB, DIEP	IDC	0	1	I	Positive	Positive	Positive	High	Dual method	Rt axilla	HTx	Not performed
8	37	Rt	BCS with SLNB	IDC	0	2	I	Negative	Negative	Positive	High	Radioisotope	Rt axilla	RTx	Not performed
9	40	Lt	BCS only	DCIS	0	0	0	Positive	Positive	Positive	Intermediate	Not performed	Lt axilla	RTx, HTx	Not performed
**Second operation**															
1	73	Rt	BCS	IDC	0	0	I	Positive	Positive	Positive	High	Dual method	Lt axilla	HTx	0/3
2	62	Rt	BCS	IDC	0	0	I	Positive	Negative	Negative	Intermediate	Radioisotope	Lt axilla	HTx	0/3
3	59	Rt	SSM only with DIEP	IDC	0	0	I	Positive	Positive	Negative	Low	Dual method	Lt axilla	HTx	0/3
4	51	Rt	BCS with SLNB	IDC	0	0	I	Positive	Positive	Negative	Intermediate	Dual method	Lt axilla	HTx	0/2
5	38	Lt	CM with SLNB	IDC	0	4	I	Positive	Positive	Positive	Intermediate	Dual method	Both axilla	HTx (+Goserelin)	0/2
6	51	Rt	CM with SLNB	IDC	0	1	I	Positive	Positive	Positive	High	Radioisotope	Both axilla	CTx, HTx	0/3
7	58	Rt	Re-excision with SLNB	IDC	1	9	IIa	Positive	Negative	Positive	High	Dual method	Both axilla	CTx, RTx, HTx	0/1
8	40	Rt	SSM with SLNB, TEI	IDC	0	2	I	Negative	Negative	Positive	High	Radioisotope	Both axilla		0/3
9	46	Lt	SSM with SLNB, DIEP flap	Mucinous ca.	0	2	I	Positive	Positive	Negative	Intermediate	Radioisotope	Both axilla	HTx	0/4

Rt, right; Lt, left; SSM, skin sparing mastectomy; DIEP, deep inferior epigastric perforator; IDC, invasive ductal carcinoma; CM, completion mastectomy; TEI, tissue expander insertion; CTx, chemotherapy; RTx, radiotherapy; HTx, hormone therapy; NACT, neoadjuvant chemotherapy.

**Table 3 jcm-09-01192-t003:** Clinical course after the first and second operations.

Case No.	Interval to IBTR (months)	F/U Duration (months)	F/U Duration after 2nd Operation (months)	Event after 2nd Operation
1	143	181	37	No
2	79	174	94	Remnant breast
3	105	139	33	No
4	91	110	19	No
5	37	76	38	No
6	33	70	36	No
7	30	68	37	No
8	28	30	1	No
9	70	75	4	No

F/U, follow up.

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
