# Peer review of "Aberrant Lymphatic Drainage in the Contralateral Axilla in Patients with Isolated Ipsilateral Breast Tumor Recurrence"

_jcm, 2020, doi:10.3390/jcm9041192_

Round 1
Reviewer 1 Report
Following minor change will improve the quality of the manuscript
Page 2 line 26 “We conducted a retrospective review of studies between 1995 and 2015”.
Please change this sentence to “We conducted a retrospective review of cases reported at Samsung Medical Center between 1995 and 2015”.
Please provide patients details in the materials and methods section for e.g.:
- What type of database used to recognize the patient population?
- Please include the number of patients used for the study and the inclusion criteria of patients selected for the study in the materials and methods. In the present format, these are mentioned in the abstract.
Author Response
Page 2 line 26 “We conducted a retrospective review of studies between 1995 and 2015”.
Please change this sentence to “We conducted a retrospective review of cases reported at Samsung Medical Center between 1995 and 2015”.
- I really appreciate for your kind review. I corrected as your comment
Please provide patients details in the materials and methods section for e.g.:
- What type of database used to recognize the patient population?
- Please include the number of patients used for the study and the inclusion criteria of patients selected for the study in the materials and methods. In the present format, these are mentioned in the abstract.
- I added in the materials and methods section as you recommended
- “Among 17,332 primary breast cancer patients were operated at SMC, and 1,032 patients with loco-regional recurrence were identified. Among them, 233 patients with isolated IBTR were identified, and 31 patients underwent re-SLNB.”

Reviewer 2 Report
Reviewer comments for “Aberrant Lymphatic Drainage in the Contralateral Axilla in Patients with Isolated Ipsilateral Breast Tumor Recurrence”
Overview:
1) Recommend minor revisions noted in detail below
2) Article trying to add to the important discussion about sentinel lymph node biopsy (SLNB) in patients receiving contralateral SLNB with ipsilateral breast tumor recurrence (IBTR) for further considerations of management and clinical implications.
Minor edit recommendations:
- In the introduction and/or discussion, please explain why contralateral SLNB surgery was indicated in patients. Presumably the isolated IBTR occurred in the contralateral breast instead of the ipsilateral breast requiring surgical resection with possible need for SLNB vs ALND vs no treatment if thought to be Stage IV disease.
- Figure 1: In the diagram, please add the 11 patients who failed re-SLNB for completeness of the schematic diagram.
- Figure 1 (Line 109): Please indicate what “a” is referencing for N=31
- Lines 118-119: Clarification needed: Did the two patients develop lymphedema or had they already had lymphedema prior to undergoing contralateral SLNB. Rationale in clarifying is to understand if lymphedema was a result of the contralateral SLNB.
- Table 1: axillary is misspelt in “No axilary surgery” listed twice
- Lines 128-129: Consider rephrasing as the current language is confusing.
- Table 2: Put in a legend for acronyms used. Please indicate what “a” is reference for Stage.
- Lines 210-214: Appears to two sentences conveying the same information. Consider revising.
- Line 211: misspelling: “patent” should be “patient”
- Consider having a native English speaker edit the language in the introduction, results and discussion.
- Page 15 (Line 216) Author contributions section is left blank. Please complete.
Author Response
Review_2
Overview:
1) Recommend minor revisions noted in detail below
2) Article trying to add to the important discussion about sentinel lymph node biopsy (SLNB) in patients receiving contralateral SLNB with ipsilateral breast tumor recurrence (IBTR) for further considerations of management and clinical implications.
Minor edit recommendations:
In the introduction and/or discussion, please explain why contralateral SLNB surgery was indicated in patients. Presumably the isolated IBTR occurred in the contralateral breast instead of the ipsilateral breast requiring surgical resection with possible need for SLNB vs ALND vs no treatment if thought to be Stage IV disease.
à I really appreciate for your valuable comment. I added the importance of contralateral SLNB in the introduction section..
à” If the patients who had IBTR identified with contralateral axillary metastasis, it can be considered as stage IV. Therefore, contralateral SLNB could be important to avoid underestimation of the status of the disease.”
- Figure 1: In the diagram, please add the 11 patients who failed re-SLNB for completeness of the schematic diagram.
à I corrected according to your recommendation.
- Figure 1 (Line 109): Please indicate what “a” is referencing for N=31
à “a” was meaning the 11 patients who failed re-SLNB. As your valuable comment, I corrected in the Figure 1 (above), and “a” was removed.
- Lines 118-119: Clarification needed: Did the two patients develop lymphedema or had they already had lymphedema prior to undergoing contralateral SLNB. Rationale in clarifying is to understand if lymphedema was a result of the contralateral SLNB.
à Two patients developed lymphedema after contralateral SLNB. I added in the result as well as in the Table 1.
- à” Two patients (22.2%) who underwent contralateral SLNB developed lymphedema after contralateral SLNB.”
- Table 1: axillary is misspelt in “No axilary surgery” listed twice
à Thank you for your detail comment. I corrected as your recommendation
- Lines 128-129: Consider rephrasing as the current language is confusing.
à I totally agree your comment. I corrected as below: “Four patients underwent only contralateral SLNB because they showed involvement of only the contralateral axilla not ipsilateral axilla on lymphoscintigraphy.”
- Table 2: Put in a legend for acronyms used. Please indicate what “a” is reference for Stage.
à “a” was typo error, so that I deleted at the table 2.
- Lines 210-214: Appears to two sentences conveying the same information. Consider revising.
- I deleted the duplicated information sentence. I really appreciate.
- Line 211: misspelling: “patent” should be “patient”
àThank you for your detail comment. I corrected as your comment
- Consider having a native English speaker edit the language in the introduction, results and discussion.
à Thank you for your recommendation. This manuscript was edited at the eWorldEditing (SKKU2004-59).
- Page 15 (Line 216) Author contributions section is left blank. Please complete.
à I added as below
à Conceptualization, Se Kyung Lee; Data curation, Jeong Eon Lee and Seok Won Kim; Methodology, Jonghan Yu; Supervision, Byung-Joo Chae; Writing – original draft, Jai Min Ryu; Writing – review & editing, Seok Jin Nam.

Reviewer 3 Report
The authors conducted a retrospective review of studies over 20 years in their hospital and identified 233 patients with isolated ipsilateral breast tumor recurrence (IBTR). Among them, 31 patients underwent repeat sentinel lymph node biopsy (SNLB); 11 ipsilateral axilla, 9 contralateral axilla, and 11 no re-SLNB. No contralateral metastasis was identified. The patients’ characteristics and prognoses were compared between ipsilateral and contralateral axillary SNLB groups in patients with isolated IBTR but there were no significant differences between the two groups except longer interval to IBTR.
Aberrant lymphatic drainage has been identified when repeat SLNB was performed in IBTR cases. The authors shared their experience of this relatively rare incidence. Aberrant lymphatic drainage could be identified to the contralateral axilla, internal mammary, and periclavicular lymph node. The authors compared only between ipsilateral and contralateral axillary SNLB groups in this study. I wonder they might identify other lymph nodes groups such as internal mammary lymph nodes. Please explain in the text whether the other aberrant lymphatic drainage groups were identified or any special reason to compare these two groups.
Lymphoscintigraphy image of the contralateral axillary SNLB may be useful to provide some visual information if any.
Two patients developed lymphedema after the contralateral axillary SNLB. Is there any assumption about correlation between lymphedema and the contralateral SNLB? Lymphedema therapists commonly shifted excess fluid in the affected limb to the contralateral axilla. Lymphatic drainage pathway of the non-edematous limb may alter to the contralateral axilla before the SNLB?
There is no description in the author contributions and require the information.
Author Response
Review 3.
- The authors conducted a retrospective review of studies over 20 years in their hospital and identified 233 patients with isolated ipsilateral breast tumor recurrence (IBTR). Among them, 31 patients underwent repeat sentinel lymph node biopsy (SNLB); 11 ipsilateral axilla, 9 contralateral axilla, and 11 no re-SLNB. No contralateral metastasis was identified. The patients’ characteristics and prognoses were compared between ipsilateral and contralateral axillary SNLB groups in patients with isolated IBTR but there were no significant differences between the two groups except longer interval to IBTR.
- Aberrant lymphatic drainage has been identified when repeat SLNB was performed in IBTR cases. The authors shared their experience of this relatively rare incidence. Aberrant lymphatic drainage could be identified to the contralateral axilla, internal mammary, and periclavicular lymph node. The authors compared only between ipsilateral and contralateral axillary SNLB groups in this study. I wonder they might identify other lymph nodes groups such as internal mammary lymph nodes. Please explain in the text whether the other aberrant lymphatic drainage groups were identified or any special reason to compare these two groups.
à Thank you for your valuable comments. At first, we experienced the patients who underwent contralateral SLNB who suffered from the blood drawing at legs to avoid lymphedema. However, there was no standard guideline because of the low incidence. We added in the introduction as below according to your comments.
- “If the patients who had IBTR identified with contralateral axillary metastasis, it can be considered as stage IV. Therefore, contralateral SLNB could be important to avoid underestimation of the status of the disease. On the other hand, patients who underwent both axillary surgeries show decreased quality of life because they should check blood pressure or take blood samples at legs to prevent higher risk of lymphedema.”
- à Also, I totally agree with your comments that it has some possibility of aberrant lymphatic drainage to the internal mammary, and periclavicular lymph node, which we did not demonstrated. I added that point in the limitations. Further study is warranted to evaluate the all aberrant lymphatic drainages. Because of the rare incidence, we are going to collect multicenter retrospective date at the Korean Breast Cancer Study Group in the future.
- Lymphoscintigraphy image of the contralateral axillary SNLB may be useful to provide some visual information if any.
à I really appreciate for your comments. I added Figure 2 and Figure 3
Fig. 2. Aberrant drainage to the contralateral axilla and not on the ipsilateral axilla
Fig. 3. Aberrant drainage to the contralateral and ipsilateral axilla
- Two patients developed lymphedema after the contralateral axillary SNLB. Is there any assumption about correlation between lymphedema and the contralateral SNLB? Lymphedema therapists commonly shifted excess fluid in the affected limb to the contralateral axilla. Lymphatic drainage pathway of the non-edematous limb may alter to the contralateral axilla before the SNLB?
à Two patients developed lymphedema after contralateral SLNB. I added in the result as well as in the Table 1.
à” Two patients (22.2%) who underwent contralateral SLNB had lymphedema after contralateral SLNB.”
à I agree with your point that lymphedema and lymphedema therapies could affect the lymphatic drainage. Unfortunately, we could not analyze the effect of lymphedema to the contralateral lymphatic drainage because it was rare disease and the limitation of retrospective design.
à I added at the limitations. “Further multicenter studies are needed to evaluate the effect of aberrant lymphatic drainages pathways.”
- There is no description in the author contributions and require the information.
à I added as below.
à Conceptualization, Se Kyung Lee; Data curation, Jeong Eon Lee and Seok Won Kim; Methodology, Jonghan Yu; Supervision, Byung-Joo Chae; Writing – original draft, Jai Min Ryu; Writing – review & editing, Seok Jin Nam.
